# Physiologically Active Molecules and Functional Properties of Soybeans in Human Health—A Current Perspective

**DOI:** 10.3390/ijms22084054

**Published:** 2021-04-14

**Authors:** Il-Sup Kim, Cheorl-Ho Kim, Woong-Suk Yang

**Affiliations:** 1Advanced Bio-resource Research Center, Kyungpook National University, Daegu 41566, Korea; 92kis@hanmail.net; 2Molecular and Cellular Glycobiology Unit, Department of Biological Sciences, SungKyunKwan University, Gyunggi-Do 16419, Korea; 3Samsung Advanced Institute of Health Science and Technology, Gyunggi-Do 16419, Korea; 4Nodaji Co., Ltd., Pohang, Gyeongsangbuk-Do 37927, Korea

**Keywords:** soybean, active molecules, soy functionality, health benefit

## Abstract

In addition to providing nutrients, food can help prevent and treat certain diseases. In particular, research on soy products has increased dramatically following their emergence as functional foods capable of improving blood circulation and intestinal regulation. In addition to their nutritional value, soybeans contain specific phytochemical substances that promote health and are a source of dietary fiber, phospholipids, isoflavones (e.g., genistein and daidzein), phenolic acids, saponins, and phytic acid, while serving as a trypsin inhibitor. These individual substances have demonstrated effectiveness in preventing chronic diseases, such as arteriosclerosis, cardiac diseases, diabetes, and senile dementia, as well as in treating cancer and suppressing osteoporosis. Furthermore, soybean can affect fibrinolytic activity, control blood pressure, and improve lipid metabolism, while eliciting antimutagenic, anticarcinogenic, and antibacterial effects. In this review, rather than to improve on the established studies on the reported nutritional qualities of soybeans, we intend to examine the physiological activities of soybeans that have recently been studied and confirm their potential as a high-functional, well-being food.

## 1. Introduction

Soybean is an excellent food resource as it contains high-quality protein, a high ratio of unsaturated fatty acids and dietary fiber, as well as other substances that possess various physiological functions [1]. Due to the recent westernization of the Korean diet, the prevalence of certain diseases, including diabetes and cardiovascular events, is increasing [2,3]. Moreover, considering that the overconsumption of animal-based food products is a contributor to the development of obesity, the functionality of soy, a plant-derived food product, is highlighted [1,4,5]. Soybean is composed of 40% protein, which is significantly higher than most other types of beans [5]. Furthermore, the high-quality protein in soy is equivalent to that found in dairy, meat, and eggs but lacks cholesterol and saturated fatty acids [1,4]. Moreover, the Food and Drug Administration (FDA) of the United States (US) recognized that the consumption of soy protein decreases the risk of cardiovascular diseases and applied the “healthy” food label, which increased public interest in soybean in the US, as well as Japan [6,7]. The consumption of soy protein by obese patients is effective in preventing and treating obesity [8,9]; not only does it inhibit fat accumulation and increase fat metabolism, but it also contributes to weight reduction by regulating the expression of appetite-suppressing factors [10]. Furthermore, the fermentation of soybean by effective bacteria, including *Bacillus* spp., produces certain enzymes and physiochemical substances that do not exist in the raw food product [11]. Soybean fermentation is accomplished by either *Bacillus subtilis* or *Bacillus licheniformis*, which weakens the putrefactive effect of intestinal bacteria, and resists pathogenic bacteria by absorbing toxic substances [12,13]. 

Soybean comprises lipids (18%), proteins (38%), carbohydrates (30%, 1:1 ratio of soluble and insoluble forms), and moisture, ash, and other substances (14%), which include vitamins and minerals that are not required in large quantities by the body (Figure 1A,B; Table 1). In relation to proteins, the nutritional value of soybean amino acids is as follows: tryptophan (1%), tyrosine (4%), valine (4%), arginine (8%), alanine (4%), aspartic acid (7%), cysteine (3%), glutamic acid (19%), glycine (4%), histidine (3%), phenylalanine (6%), isoleucine (5%), lysine (8%), leucine (8%), methionine (1%), proline (5%), serine (5%), and threonine (4%) (Figure 1C). Soybean is high in fiber, protein, and phytoestrogens, low in saturated fats, free cholesterol, and lactose, and a good source of omega-3 fatty acids and antioxidants. Moreover, the stable storage of soybean can be improved by the addition of 12–14% moisture [1,4]. Recently, much attention has been paid to soybean as a functional food because several studies have shown that it contains at least 14 beneficial phytochemical substances, including phytic acid, triterpenes, phenolics, flavonoids, lignans, carotenoids, and coumarins, as well as protease inhibitors, oligosaccharides, and dietary fibers [1,4,5,14,15]. These compounds have purported anticancer, antiaging, antirenal failure, antiobesity, and anticholesterolemic properties, while also being shown to inhibit HIV, and prevent gallstone formation, senile dementia, and hyperlipidemia. Furthermore, soybean promotes diuretic action, suppresses arteriosclerosis, provides relief from constipation, and prevents cardiovascular diseases (Figure 2) [15,16,17,18,19]. Soybean also contains substances that are involved in intestinal regulation, have antioxidative properties, prevent osteoporosis, lower blood pressure, have antithrombotic effects, boost immunity, and promote liver functions, and therefore, can be inferred to be closely related to the prevention of certain chronic diseases [15,16,17,18,19,20,21]. 

Interestingly, certain compounds in soybean, such as lectin and trypsin inhibitors, were originally reported as harmful substances; however, new physiological functions, including the prevention and improvement of diabetes and antitumor activity, have been identified [22,23]. Moreover, phospholipids, which are also abundant in soybean, reduce plasma lipid and total cholesterol levels [24]. Soybean oligosaccharides have the ability to promote the proliferation of probiotic bacteria, such as *Bifidobacterium* and *Lactobacillus* spp. [25,26]. Additionally, the low molecular weight compounds in soybean have demonstrated various functions [5,27]. For instance, phytic acid, an inositol with six attached phosphates, saponin, and isoflavone, which are primarily responsible for the bitter taste of soybean, have also exhibited antitumor and antioxidant properties [15,16,17,18,19,20,21]. The primary aim of this review is to highlight the novel functionality of various soybean-derived molecules that have recently received immense attention. 

## 2. Phenolic Compounds

Most plants contain antioxidants, referred to as phenolic compounds (or phenolics), that include flavonoids, hydroxycinnamic acid derivatives, phenolic acids, and tannic acid [28]. Tannic acid is a naturally occurring polyhydroxyl phenol ester of gallic acid [29]. In addition, soybeans contain isoflavone, a derivative of phenolic acids and flavonoids [30].

### 2.1. Phenolic Acids

Soybeans contain eight phenolic acids, namely, *p*-hydroxy benzoic acid, chlorogenic acid, cinnamic acid, ferulic acid, gentisic acid, salicylic acid, syringic acid, and vanillic acid (Figure 3) [31,32]. Chlorogenic acid is hydrolyzed to form caffeic acid, both of which cause browning, a harmful effect in food, that leads to nutrient loss and affects color and flavor [33]. The removal of phenolics with activated carbon has been shown to increase flavor and improve digestibility in vitro [34]. Conversely, caffeic acid and chlorogenic acid also have the potential to block nitrosamine genesis both in vitro and in vivo [35]. In addition, these phenolics inhibit the metabolism of aflatoxin B1 in rat liver [36]. Phenolic acid can also act as an antioxidant to inhibit DNA damage caused by reactive oxygen species [37]. 

### 2.2. Isoflavones

Isoflavones are phytoestrogens which, along with lignans, are found in plants [30]. These are phytochemicals that have physiological activity similar to estrogen and are activated by the intestinal flora [12,30]. Soybean hypocotyls contain many isoflavones that are classified into four categories based on their chemical structures: (i) aglycons including daidzein, genistein, and glycitein; (ii) glycosides including daidzin, genistin, and glycitin; (iii) three types of acetyl glycosides; and (iv) three types of manonyl glycosides (Figure 4) [12,38]. According to several studies, isoflavones have the ability to activate estrogen receptors in the vagina, oocytes, and mammary glands, and can possess estrogen or antiestrogen properties depending on the physiological environment or their chemical structure [39,40]. For instance, isoflavone is an antiestrogen that reduces the risk of breast and prostate cancers [41,42], and also has antioxidant effects similar to vitamin E and C in vivo and in vitro [43,44]. Furthermore, isoflavone is produced by an oncogene and functions as an inhibitor of tyrosine protein kinase [45]. Among the soybean isoflavones, genistein inhibits the growth of cells that cause breast, colon, lung, prostate, and skin cancers in vitro (Figure 5) [36,46]. In addition, genistein inhibits the formation of boils by preventing vasculogenesis, thus blocking the supply of oxygen or nutrients [47]. 

Additionally, studies investigating the effects of isoflavone ingestion in Western women have shown that isoflavones affect the menstrual cycle, such that it reduces the risk of breast cancer [48]. Furthermore, isoflavones can have weak estrogenic effect and can reduce the severity of symptoms associated with menopause, without eliciting any negative side effects [30,49]. Isoflavones reduce blood cholesterol by as much as 35%, suggesting their potential as cholesterol-lowering agents [50], while casein, an animal protein, has been found to increase blood cholesterol [51]. In people with insufficient protein intake, protein is generated from accumulated fat [1]. Since the required fat is transported via blood vessels, there is a corresponding increase in fat levels in the blood, which consequently increases blood cholesterol [52]. Thus, consuming high-quality soy protein helps to lower blood cholesterol.

In the United States, approximately 15% of women who have entered menopause receive estrogen administration as hormone therapy [53]. However, estrogen administration increases the likelihood of cancer in reproductive organs [54]; thus, soybean, a natural food, is gradually being considered as a potential substitute for estrogen in this population [6]. Estrogen also lowers the risk of osteoporosis by promoting vitamin D activity, which prevents calcium elution of bones and increases calcium absorption [55]. Specifically, the isoflavones in soybeans are structurally and functionally similar to estrogen, which is why they are also referred to as phytoestrogens [30]. In fact, isoflavones have been highlighted as a potential source of estrogen that does not elicit negative side effects [30]. In addition, isoflavones exert excellent anticancer effects; most of the anticancer effects of isoflavones are produced by genistein [30,56]. Although genistein has been primarily investigated in breast cancer, it has been shown to weakly bind estrogen receptors and promote normal cell division while repressing cancer cell division [57]. Isoflavones also alleviate hot flashes associated with menopause without inducing hyperlipidemia or altering the muscle layers in the breast and uterus, which are common side effects observed following estrogen administration [49,58]. Furthermore, isoflavones prevent bone reabsorption and increase bone density to prevent osteoporosis, which is common in older women [20,59]. The physiological roles of isoflavones are summarized in Figure 6.

## 3. Phytic Acid

Phytic acid, also called myo-inositol hexaphosphate (IP6), consists of a myo-inositol ring and six symmetrically attached phosphate groups [60]. Although phytic acid is present in plants, it is especially common in grains and legumes at 0.4–6.8%, while soybean seeds contain 2.58% phytic acid [61,62]. During food processing, phytic acid is hydrolyzed and decomposed into IP1, IP2, and IP3 (containing one, two, and three phosphate groups, respectively), which are myo-inositols bound to fewer phosphate groups [61]. In humans, 1–3% of the total dietary phytic acid is excreted in the urine at approximately a 0.5–0.6 mg/L concentration [63]. Phytic acid, which is widely distributed in the outer shell of beans and grains, forms chelates with divalent ions, such as Ca^2+^, Mg^2+^, Zn^2+^, and Fe^2+^, making absorption into the small intestine difficult [61,64]. Phytic acid also interferes with the utilization of minerals, acting as a non-nutritional component, and hinders the action of important digestive enzymes such as pepsin, trypsin, and α-amylase by strongly binding to the protein base [65,66]. Hence, phytic acid is considered a non-nutritional compound as it can affect the utilization of minerals in the body by binding to them and reducing their absorption [67]. However, it has recently gained favor due to its recently described antioxidant, anticancer, and lipid-lowering effects [68].

Phytic acid has several physiologically active functions, including storage of phosphorus and cations [69]. Iron is known to cause oxidative damage in vivo. The hydroxyl radicals produced by iron cause oxidative damage by inducing the oxidation of cells or lipids [70]. Meanwhile, phytic acid can bind to iron to inhibit the production of hydroxyl radicals, thereby preventing the oxidation of cells [67,68]. This function has also been highlighted in the context of food processing; research is being conducted on the use of phytic acid supplementation to suppress oxidation that may occur during food processing [61,68]. Moreover, people with a higher intake of grains and vegetables containing large amounts of phytic acid have a lower incidence of colorectal cancer, due to phytic acid activating the expression of tumor-suppressor genes (e.g., p53 and WAF-1/p21) [71,72]. Phytic acid also exhibits antitumor activity by reducing cancer cell proliferation and increasing cell differentiation (Figure 7) [71]. In addition, lower inositol phosphates, such as IP3 and IP4 (containing three and four phosphorus groups, respectively) play an important biological role in regulating cell-to-cell responses and are known to act on the signaling systems in the body [73].

## 4. Protease Inhibitors

Protease inhibitors (PIs) are found in soybeans and other plant systems, including grains, grass, potatoes, fruits, vegetables, peanuts, and corn [74]. Kunitz and Bowman–Birk types of PIs are found in soybeans and inhibit the activity of chymotrypsin, elastase, and serine proteases [75]. For the past 40 years, the PIs of soybeans have been discussed primarily as antinutritional inhibitory factors; however, more recently, they have also been highlighted for their apparent anticancer properties [76]. The mechanism underlying the health benefits of PIs is centered on their antioxidant activity, as trypsin inhibitors have been shown to block the generation of free radicals, thereby preventing cells from being transformed by oxidative damage [74]. For instance, the Bowman–Birk type PI, which has a chymotrypsin inhibitory effect, inhibits the expression of the oncogene *MYC*, which encodes c-MYC, reduces the production of hydrogen peroxide, an oxygen radical in the body, and prevents the destruction of DNA’s helical structure and DNA oxidation by inhibiting the function of the tumor promotor 12-*o*-tetradecanoylphorbal-13-acetate [76,77]. According to a recent study, not only the PIs of soybeans, but also retinoids, garlic acid, epigallocatechin gallate, nicotinic acid, and tamoxifen of some plants also function as cancer-preventing agents that inhibit the production of superoxide radicals or H_2_O_2_ by tumor promotor factors, despite their structural differences [78]. Furthermore, trypsin inhibitors in soybeans promote insulin secretion, thus normalizing blood sugar levels (Figure 8) [74,79].

## 5. Lignans

Lignans are present in plants in small quantities and participate in the construction of the cell wall framework when bound [80,81]. When ingested, they are converted to enterodiol or enterolactone by enteric bacteria and are subsequently excreted in urine in the form of glucuronide conjugates [81,82]. The lignan content in grains is generally from 2–7 mg/kg, and varies depending on the type of grain [83]. For instance, flax seeds and soybeans are rich in lignans or lignan precursors. Lignans have properties similar to estrogen due to their similar chemical structures, and are also designated as phytoestrogens as they are capable of regulating estrogen levels [84]. In fact, studies have shown that ingesting large amounts of lignans may lower the body’s free estrogen content [85]. Accordingly, regular consumption of foods rich in lignan precursors may reduce the risk of breast cancer caused by estrogen [84]. Indeed, a study has suggested that lignin downregulates the proliferative potential of breast cancer cells in a tissue culture system [86]. Specifically, lignans inhibit the activity of 5α-reductase and 17β-hydroxysteroid dehydrogenase [87], which are involved in the biosynthesis and metabolism of estrogen, or inhibit 7-α-hydroxylase activity, which is involved in the formation of bile acids from cholesterol [88], thereby potentially lowering the risk of sex hormone-related cancer and colon cancer, respectively [89,90]. Furthermore, the ingestion of foods high in lignans can enhance anticancer properties through synergistic effects with flavonoids and other phytochemicals (Figure 9) [84,91,92].

## 6. Saponins

Soybeans have the highest saponin content among all edible legumes [93]. Saponins can be categorized as either steroid or triterpene saponins according to the chemical properties of the covalently attached non-saccharide [94,95]. Triterpene saponin can be further classified into oleanane, ursane, dammarane, and cycloartane according to the skeleton of the non-saccharide segment [96]. Soyasaponins are classified into group A (soyasapogenol A), group B (soyasapogenol B), or group E (soyasapogenol E) according to the non-saccharide segment (Figure 10A) [97]. A total of 11 different saponins, six from group A and five from group B, can be isolated from the hypocotyl of soybean [98]. The monosaccharides of soyasaponins are D-galactose, D-glucose, L-arabinose, L-rhamnose, D-xylose, and D-glucuronic acid. Saponins are most common in the germ layer and are also found in the hypocotyl, but are not present in the outer skin [99]. The type and content of soyasaponins vary from species to species [97]. Group A saponins can be isolated from the hypocotyl at levels of 0.36–0.41%, while group B ranges from 0.26–2.75% [97,100]. Additionally, group B saponin content increases during germination [101]. Although there have only been a few studies on the changes in saponin content during the cooking or processing of soybeans, its abundance is reduced during fermentation by enzymes from microorganisms [4,102]. 

Saponin is a bipolar, heat-stable sugar complex that was previously known as a non-nutritional substance with a bitter and stringent taste [103]. However, recent research has revealed that it possesses physiologically active functions such as lowering cholesterol, stimulating immune responses, and anticancer effects, resulting in it being spotlighted as a functional nutrient [104,105,106,107]. Specifically, soybean saponin can reduce the time required for harmful substances to contact the mesentery, thus facilitating more rapid absorption of these harmful components, effectively weakening their toxicity [105,108]. Since saponin shares a similar chemical structure with cholesterol, it also inhibits cholesterol absorption, thereby increasing its release [52]. In addition, the synergistic effect of saponin and vitamin E (tocopherol) prevents skin blemishes and facilitates blood circulation [109]. Vitamin E not only reduces the level of low-density lipoprotein (LDL), frequently called as “bad cholesterol”, in the circulation and lowers blood viscosity to help blood flow more smoothly, but also prevents the formation of brown spots (also known as age spots) that form on the face of middle-aged and elderly people [110,111,112].

Additionally, saponins, such as phytic acids, function as antioxidants, thereby inhibiting cell damage caused by free radicals [16]. Saponins can also suppress the rate of DNA mutation, which prevents colon cancer in particular [113]. Soybean saponins have a chemical structure similar to licorice saponins, and their function as an anticancer agent is being investigated [114]. Specifically, the role of saponin as an enhancer of killer cell activity, as a sarcoma-specific cell toxin, in the inhibition of DNA synthesis in tumor cells, and in the reduction of cervical and epidermal cancer cell growth has been reported (Figure 10B) [95,115,116]. Recently, group B saponins extracted from soybeans have been shown to inhibit HIV infection [117].

## 7. Dietary Fiber and Soy Oligosaccharides

Soybean is rich in dietary fiber, a food component that cannot be broken down by digestive enzymes in the body. Dietary fibers can be classified into water-soluble fibers, such as pectin and gum, and water-insoluble fibers, such as cellulose and lignin [118]. Water-soluble dietary fiber is fermented by colonic microorganisms and is involved in the production of short-chain fatty acids, including acetic acid, butyric acid, and propionic acid, which are all major nutrients of colonic cells, and function in cholesterol absorption [119,120]. Water-insoluble dietary fiber is effective in preventing constipation as it increases bowel movements by enhancing intestinal function [121]. Soybean shells contain large quantities of water-soluble fibers, while insoluble fibers and pectin make up the cell wall [118,122]. The properties of dietary fiber include water retention, swelling, organic molecule absorption, ion absorption and exchange, and decomposition by intestinal microorganisms, so it acts alone or in combination to elicit several physiological activities [118,123]. In particular, one of the most important effects of soybean dietary fiber is the lowering of cholesterol levels [52]. Furthermore, soybean fiber plays an important role in normalizing bowel movement, is involved in controlling constipation, and reduces the intestinal transit time of food [124].

Soybean oligosaccharide is a generic term for the soluble oligosaccharides in soybeans that contain approximately 4% stachyose and 1% raffinose (Figure 11A) [125]. They are not abundant when the plant is immature, but levels rapidly increase during maturity [125,126]. Soybean oligosaccharides are indigestible, meaning they are not digested or absorbed as a nutrient in humans, but are instead considered as a flatulence factor that induces production of gases, such as CO_2_ or methane, by the flora in the large intestine [126,127]. However, they have recently received increasing attention due to their promotion of beneficial bacterial growth in the intestines [107,127]. Moreover, soybean oligosaccharides and dietary fiber promote vitamin synthesis in the intestine, inhibit the growth of harmful and external bacteria, and inhibit the production of ammonia and amines [123,128]. In addition, they act as a growth promotor of *Bifidobacterium*, a useful bacterium that acts as an anti-inflammatory agent by enhancing immune function and promoting peristaltic movement of the intestine, while aiding in digestion and absorption [127,129]. *Bifidobacterium* produces lactic acid to maintain the intestinal pH, thereby suppressing the growth of harmful bacteria and improving bowel movement to help prevent constipation or deterioration of intestinal function [130,131]. In addition, it functions to prevent the absorption of ammonia and H_2_S, which are harmful substances in the intestines, suppress the production of carcinogens such as indole, skatole, and phenol, and reduce the effect of insulin resistance and cholesterol concentration on high blood pressure (Figure 11B) [123,132].

## 8. Soy Proteins and Peptides

Soy protein contains most of the protein types necessary for humans, making soybeans a major source of vegetable protein, providing cheap, high-quality proteins, and essential amino acids [1,5,133]. Soy protein is used in various foods, such as baby food, sports drinks, milk or meat substitutes, and grain-fortified foods to provide physicochemical functions [108,134]. In fact, soy protein-based diets have recently received attention as a biological response modifier for heart disease, obesity, cancer, and diabetes [5,52,135,136]. Serum cholesterol levels are highly correlated with the development of atherosclerosis, while dietary proteins and lipids affect serum cholesterol levels. In particular, compared to animal protein, the consumption of vegetable proteins, including soy proteins, is more effective at lowering the concentration of serum cholesterol, highlighting the benefit of consuming a well-balanced protein diet [5,52,137]. Recently, the prevalence of diseases caused by obesity, such as diabetes and cardiovascular diseases, has increased due to the influence of Western diets [136]. Excessive consumption of animal foods has been raised as a primary cause of obesity, and the functionality of soybeans, a vegetable food, has emerged to combat this [5,52]. Soybeans contain approximately 40% protein, and soy protein inhibits fat accumulation, increases fat metabolism, and controls the expression of appetite suppressors, thereby contributing to weight control [9,52].

Soybean supplies the majority of the protein content that humans consume and is a leading source of high-quality and essential amino acids derived from plants [138]. Although there have been numerous studies discussing the technical use of soy protein since the 1950s, the consumption of the protein has yet to draw level with its production [139]. Soy protein is used in foods of physiological functionalities like infant formulas, sports beverages, dairy products, meat substitutes, and fortified grain products [4]. Soy protein has recently received attention as a phytochemical substance for its significance in cardiovascular diseases, obesity, cancer, and diabetes [5,52,140,141,142]. Serum cholesterol level is significantly correlated with the onset of arteriosclerosis, and lipids as well as dietary protein affect the serum cholesterol level [142,143,144]. Compared to animal-based protein, plant-derived protein is particularly effective in lowering the serum cholesterol level, and thus a balanced consumption of both types of protein is recommended. Among the other sources of plant-derived protein, soy protein especially manifests a cholesterol-lowering effect [5,52,137].

Peptides are a combination of varied amino acid chains that form polymeric low molecular amino acids of less than 10,000 Da [5]. Peptides are involved in supplying nutrition, the sensory function of taste, solubility, and emulsifiability, as well as various physiological activities, including anticancer properties, lowering of blood pressure and serum cholesterol levels, strengthening of immunity, and promotion of calcium absorption [5,14]. Soy protein is partially hydrolyzed into soy peptides through enzymatic reactions. The soy amino acids are absorbed by the body well and hydrophobic amino acids are broken down at their C or N terminals, lessening their bitter taste [145,146]. Furthermore, soy peptides exhibit strong gel-forming abilities, which fortify their emulsifying and foaming functions [147,148]. By checking the reabsorption of bile acids within digestive organs, soy peptides (i.e., lactostatin (IIAEK)) can effectively lower blood cholesterol levels, thus effectively decreasing LDL and fat accumulation (Figure 12), which enables their use in preventing and treating arteriosclerosis [5]. Soy proteins also contain angiotensin-I converting enzyme (ACE-I)-inhibiting peptides that decrease blood pressure levels (Figure 13), as well as antithrombotic peptides that inhibit platelet aggregation [149,150]. The diverse effects of soybean-derived bioactive peptides are summarized in Table 2. Indeed, new functional foods that contain soy peptides, capable of eliciting physiological activities, have recently been commercialized and made available for applications in health-oriented foods, functional foods, pharmaceuticals, and cosmetics [5].

## 9. Lecithin

Lecithins are complex lipids that are abundantly present in egg yolk, soybean oil, liver, and brain and form a lipid or lipid protein in fat spheres [151]. They are composed of a fatty acid chain on one side, with strong lipophilicity, and a phosphoric acid and choline moiety on the other, with strong hydrophilicity; thus, they are widely used as an emulsifier to stabilize mixtures of water and oil [151,152]. Moreover, lecithin is commonly used as an antiscattering agent as well as a humectant, to reduce viscosity and control crystallinity [152,153]. The term lecithin, in its natural state, refers to various phospholipid mixtures, such as phosphatidylcholine, phosphatidylethanolamine, and phosphatidylinositol (Figure 14, left panel), which are collectively called lecithin, however, chemically, lecithin refers to phosphatidylcholine [154,155]. Egg yolk lecithin refers to lecithin extracted from eggs, whereas soybean lecithin refers to that extracted from soybeans [156]. The difference between soybean lecithin and egg yolk lecithin primarily lies in the differences in phospholipid and fatty acid composition [157]. Soybean lecithin contains equal proportions of phosphatidylcholine, phosphatidylethanolamine, and phosphatidylinositol, whereas egg yolk lecithin contains approximately 70% phosphatidylcholine, low levels of phosphatidylinositol, and some sphingomyelin [158,159]. 

Phospholipids in vivo are not only important components of cell membranes but are also distributed throughout tissues and organ systems, thus playing an important role in physiological functions (Figure 14, right panel). Consequently, lecithin is involved in a myriad of metabolic processes, including absorption of fat-soluble nutrients and vitamins, as well as discharge of waste products [160,161]. It is also involved in solubilization of cholesterol to reduce blood cholesterol [162,163]. In addition, it is effective in preventing diabetes, maintaining kidney function, normalizing liver function, and improving digestibility [4,40,164]. In fact, supplementing lecithin in a low-fat, low-cholesterol diet has been shown to lower LDL cholesterol by 15% compared to that by low-fat diet alone, while significantly increasing HDL cholesterol levels [165]. Considering that lecithin dissolves, washes, and transports fats in the body, it is beneficial for skin health as it not only removes triglycerides, but also waste and oily substances in blood vessels [157]. It is also used as an antioxidant to minimize oxidative damage of vitamin A [166].

Lecithin has positive effects by improving brain function and preventing senile dementia [167]. Moreover, lecithin in soybeans effectively prevents the reduction of acetylcholine in the brain [168]. For instance, one study showed an increase in the amount of acetylcholine in the brains of rats that were administered lecithin [169]. Increased activity of the cerebrum increases the consumption of acetylcholine [170]. Phosphatidyl choline affects lipid metabolism, fat absorption, and nerve function, while phosphatidyl inositol is involved in hormone expression, cell proliferation, cell division, and liver metabolism [171,172]. On the other hand, choline, the base constituent of phosphatidyl choline, is a precursor to acetylcholine, which prevents amnesia [173].

## 10. Conjugated Linoleic Acid

Conjugated linoleic acid (CLA) is a group of unsaturated fatty acid derivatives that are named according to the position and geometric isomers of CLA (e.g., 9-*cis*, 12-*cis*-octadecadienoic acid) [174]. Linoleic acid has two double bonds, thus, a total of eight isomers are naturally present, accounting for more than 98% of all CLA isomers [175]. All of these isomers are assumed to be trans fatty acids and are not nutritionally beneficial. CLA was first isolated from fried ground beef, and has attracted attention as a potential anticancer agent as it inhibits the development of skin cancer in mice [174]. Since then, the ability of CLA to suppress cancers, including breast and colon cancers, has been demonstrated through animal studies with cancers induced by various types of chemical carcinogens [176]. Of the eight CLA isomers, 9-*cis*, 11-*trans*-octadecadienoic acid exerts a strong anticancer action [176,177]. Moreover, CLA exhibits an antioxidant effect that is stronger than that of α-tocopherol and similar to that of butylated hydroxy toluene [178]. The antioxidant effect of CLA is expected to exert an anticancer activity by protecting the cell membrane from free radicals [175,179]. In addition, CLA has an inhibitory effect on atherosclerosis by significantly lowering total cholesterol, LDL cholesterol, and triglycerides in the blood, thereby effectively reducing the development of atherosclerotic plaques [180,181]. Furthermore, supplementing CLA in livestock feed promotes growth and improves feed efficiency by reducing body fat and increasing the amount of lean meat [182]. Meanwhile, CLA in the meat or milk of livestock is produced by conversion of linoleic acid into CLA by commensal microorganisms, especially intestinal bacteria [183,184]. Activation of CLA-mediated *CYP7A* was associated with the regulation of adipocyte differentiation, insulin resistance, lipid metabolism, carcinogenesis, inflammation, and immune functions (Figure 15).

## 11. Pinitol

Pinitol (D-Pinitol, 3-*O*-methyl-D-chiro-inositol) is a naturally occurring blood sugar regulator found in legumes and pine needles and is an active ingredient in many plants used as folk remedies for diabetes in numerous countries around the world. Chiro-inositol, a structural isomer of myo-inositol, is a compound in which a methyl group is bound to the 3rd carbon by an ether bond [185,186,187,188]. Pinitol is converted to chiro-inositol by the removal of the methyl group attached to the 3rd carbon under the influence of gastric acid [188,189]. Chiro-inositol is then absorbed into the blood vessels where it triggers galactosamine and insulin signaling to participate in normal energy metabolism [190,191]. The concentration of chiro-inositol in the body is low in diabetic patients with impaired glucose tolerance or insulin resistance, while administration of artificial chiro-inositol has been shown to improve insulin resistance, suggesting improved sugar metabolism and blood sugar regulation [192,193]. Hence, ingested pinitol may have similar effects to chiro-inositol in the normalization of sugar metabolism [194]. Indeed, pinitol has previously been used as an oral hypoglycemic agent to control blood sugar in patients with type 2 diabetes [195].

D-Pinitol (DP) has been shown to prevent diabetes-induced endothelial rupture in the cardiovascular arterial vessel [196]. The preventive activity has been attributed to its antioxidative effect on nitric oxide-mediated signaling [196]. However, the mechanistic explanation how DP exerts antihyperglycemic functions is unclear [196]. Figure 16 illustrates the presumable antihyperglycemic activity of DP. Regarding DP-associated pharmacology, several plant remedies are projected to gain a huge amount of interest owing to their pharmaco-biological functionalities toward anti-inflammatory and antioxidative features. The multiple in vitro and in vivo functions exerted by pinitol can reduce and prevent inflammatory and oxidative circumstances [196].

## 12. Conclusions

Soybeans contain high-quality proteins and are a nutritionally superior food resource containing high unsaturated fatty acid ratios and high levels of dietary fiber. In addition to the excellent nutritional properties of soybeans, their value has increased recently as it has been reported that substances previously known as antinutritional factors can exert anticancer actions as well as several other physiological functions. Soybeans have been reported to contain various physiologically active substances including soy protein, oligosaccharides, dietary fiber, isoflavone, saponin, lecithin, phytic acid, protease inhibitors, and pinitol. In addition, soybeans contain more vitamin B_1_ and E than other grains. They are also a desirable food in terms of nutrients, containing both water-soluble and fat-soluble vitamins. They are also a good source of vitamin C as the oligosaccharides in soybeans can be reduced to vitamin C. Moreover, vitamin A, a fat-soluble vitamin, exists in the form of beta carotene. Vitamin E, or tocopherol, functions as an antioxidant to prevent aging. Asparagine in soybeans also helps to relieve hangovers by removing oxides formed during metabolism of highly toxic alcohols. Recently, the benefits of soybeans have surpassed their nutritional value to highlight their functional role in the prevention and treatment of diseases, and as a result, research on soybeans is rapidly growing. Further studies are necessary for the discovery of new functional substances of soybean and to investigate the various underlying mechanisms associated with them.

## 13. Perspectives

This review aims to discuss the characteristics of the unique soybean-derived bioactive substances and its potential as a highly functional food, rather than the plants produced with the latest technology to improve crop production and environmental adaptation. Over the past 50 years, research on crop and food improvement through genetic modification, especially in soybean, has been actively conducted. Currently, with the global advances in the field of biotechnology related to agriculture and food, the research studies in the development of new crops and new varieties are moving toward a strategy to combine molecular breeding with traditional breeding [197]. Genetically modified (GM) crops refer to those produced by a technique that modifies the genetic material of a host plant by combining it with the genetic materials derived from other plant species or its own, to increase its productivity following better environmental adaptation [198]. Such genetic modification technologies have been widely used in recent years, especially due to the advantages of improving productivity and resistance to environmental stress and disease in a short time [199]. In other words, the importance of genetic engineering technology is increasing as it can effectively create new crops to cope with the current climate change problem and global population growth [200]. However, the concerns, including controversies, over its safety for human consumption and impact on the environment, and popular prejudices against GM crops and many unscientific, irrational regulations, need to be addressed to successfully implement GM crops as food [201,202,203].

Furthermore, with the advancement of technologies, several studies reporting many GM crops have been documented worldwide. In particular, GM soybeans are cultivated in many parts of the world, and products obtained from such GM crops are either served on our table in various forms of food or are integrated with processed food items [199]. Thus far, there have been divided opinions on how these GM crops and foods will affect our health; with some warning of their adverse effects on the environment and humans and requesting special regulations to ensure their minimization [203]. In some European countries, more stringent legal regulations govern the production and import of GM crops [204]. Therefore, in recent years, more advanced genetic engineering technologies, including genome editing [205], are being applied to develop new methods to correct genes without using genetic modification technology. These new technologies are combined with various advanced biotechnological tools to create a new generation of genetically engineered crops [199]. The increasing rate of population growth foreseen in the near future and other global problems such as climate change, food shortage, and environmental pollution indicate that the traditional breeding techniques are inadequate to meet the ever-growing demand. Therefore, the convergence of biotechnology, genetic engineering, and molecular breeding technologies with traditional crop breeding is expected. Furthermore, it can also solve the difficulties in the development of highly functional health foods.

## Figures and Tables

**Figure 1 ijms-22-04054-f001:**
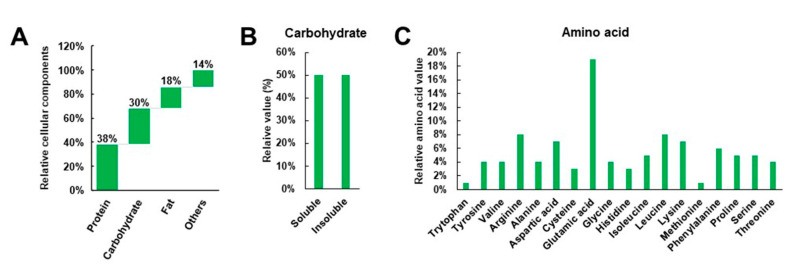
Nutritional value of carbohydrates, proteins, and fats (**A**), ratio of soluble and insoluble forms of carbohydrate (**B**), and composition of protein-derived amino acids (**C**) in soybeans [1].

**Figure 2 ijms-22-04054-f002:**
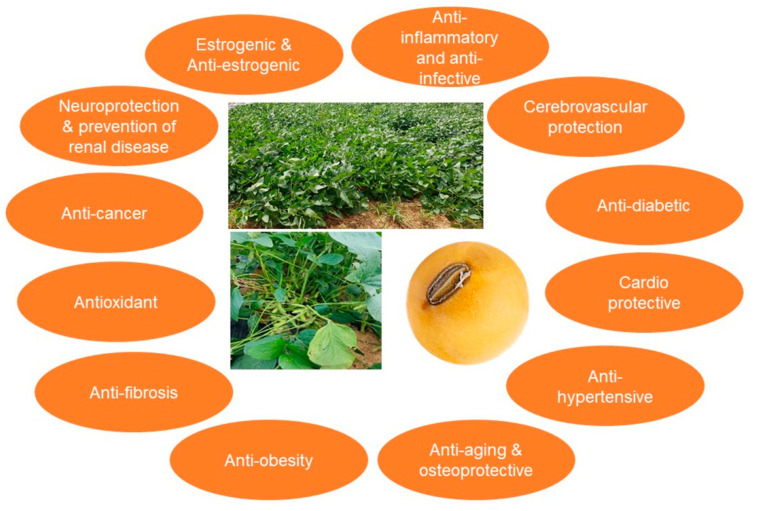
Potential beneficial health effects of soybean molecules.

**Figure 3 ijms-22-04054-f003:**
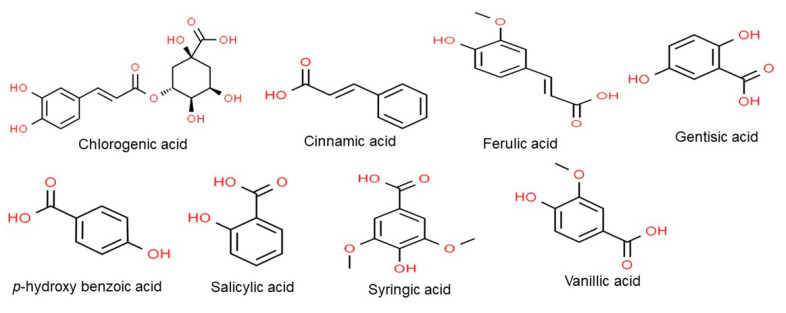
Chemical structure of the major classes of phenolic acids. Structures were drawn using Chem Spider tool.

**Figure 4 ijms-22-04054-f004:**
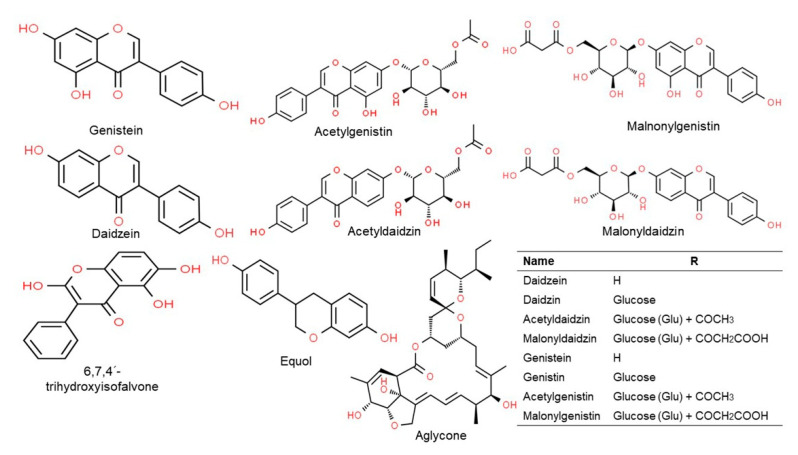
Chemical structure of the major classes of isoflavones. Structures were drawn using Chem Spider tool.

**Figure 5 ijms-22-04054-f005:**
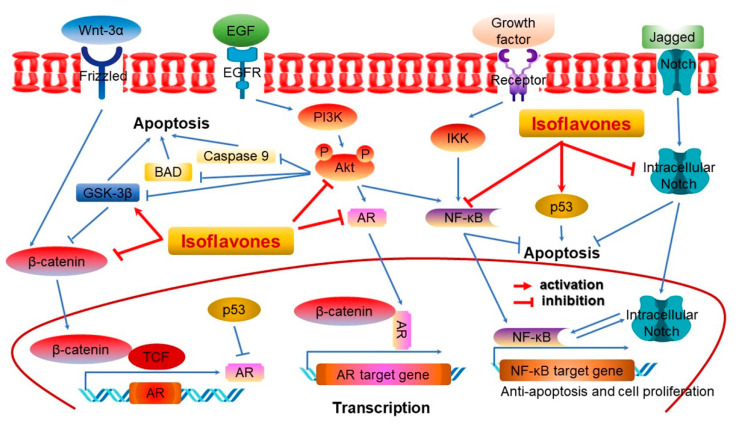
Schematic of multiple signaling pathways involved in isoflavone-induced cancer cell death [30]. Wnt-3α, a protein of the Wnt family, plays critical roles in regulating pleiotropic cellular functions [30]. AR, androgen receptor; Akt, a serine/threonine-specific protein kinase known as a protein kinase B; BAD, Bcl2-associated agonist of cell death; GSK-3β, glycogen synthase kinase 3 beta; EGF, epidermal growth factor; EGFR, epidermal growth factor receptor; TCF, T-cell specific transcription factor; IKK, IκB kinase; p53, tumor protein-53; β-catenin, a core component of the cadherin protein complex; PI3K, phosphatidylinositol-3-kinase; NF-κB, nuclear factor kappa light-chain-enhancer of activated B cells; Notch, a family of type-1 transmembrane proteins; →, activation; ⊥, inactivation. Figure adapted from Li, Y. et al. [30].

**Figure 6 ijms-22-04054-f006:**
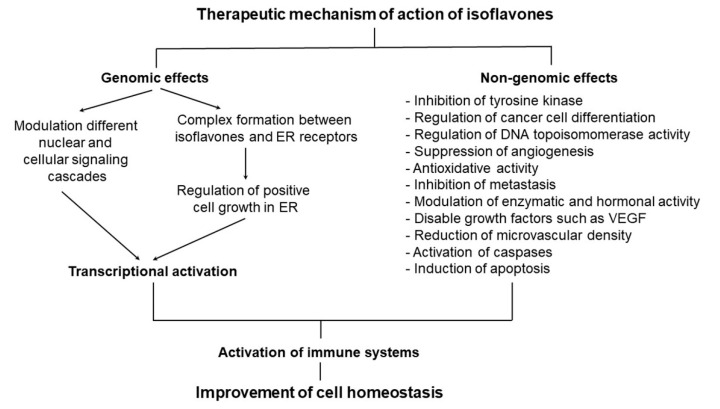
Mechanism of action of soybean isoflavones [12,30]. ER, endoplasmic reticulum; VEGF, vascular endothelial growth factor.

**Figure 7 ijms-22-04054-f007:**
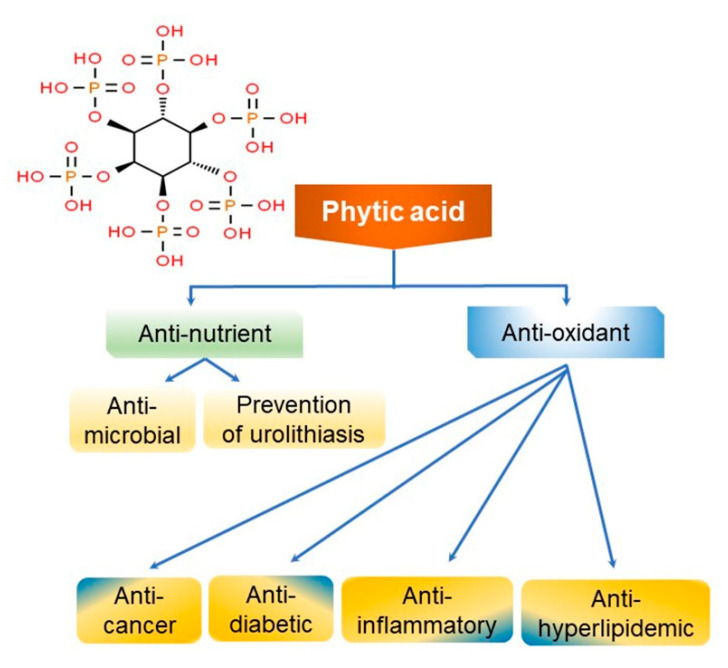
Structure and schematic representation of clinical properties of phytic acid [68,71].

**Figure 8 ijms-22-04054-f008:**
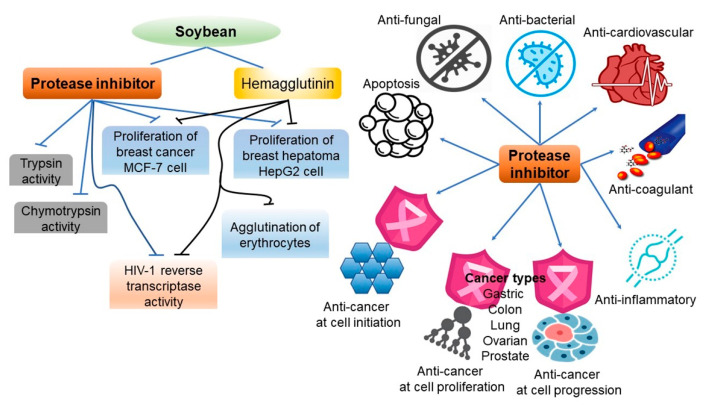
Biological activities of protease inhibitors (**left panel**) and schematic representing major pharmacological activities of soybean (**right panel**). Figure adapted from Srikanth, S. et al. [74].

**Figure 9 ijms-22-04054-f009:**
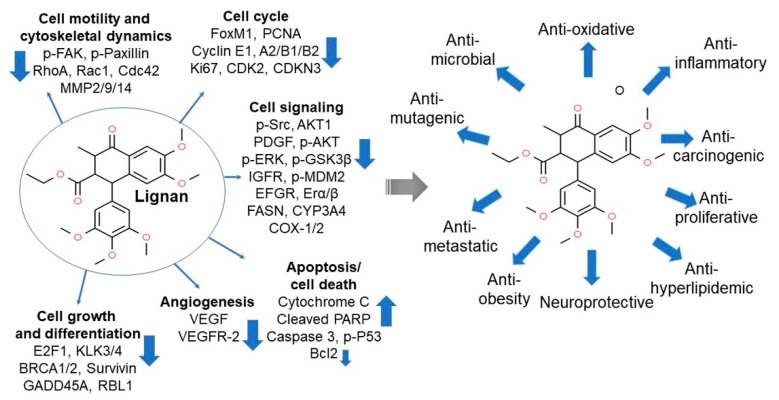
The cellular and molecular targets for anticancer (**left panel**) and protective health benefits of lignan (**right panel**). Cancer metastatic potentials are downregulated by blocking the cytoskeleton-driven cell mobility precursors. Regulation of cell differentiation and proliferation and cell cycle arrest can also interfere with survival and growth of cancer cells. Antiangiogenic starvation of tumor cells and apoptotic induction contribute to tumor progression, survival, and invasion potentials. Inhibiting various cellular signaling pathways associated with downstream intracellular kinases, including AKT, ERK, and mitogen kinases, regulates cellular metabolic pathways, triggering the suppressed tumor growth and progression [91]. In addition, lignan-containing diets or supplements can enhance general health and prevent various diseases. Cdc42, cell division control protein 42 homolog; AKT1, RAC-alpha serine/threonine-protein kinase; p-FAK, phosphorylated focal adhesion kinase; GADD45A, growth arrest and DNA damage inducible alpha; IGFR, insulin-like growth factor 1 receptor; p-ERK, phosphorylated extracellular-signal-regulated kinase; p-paxillin, phosphorylated focal adhesion-associated adaptor protein; RhoA, Ras homolog family member A; Rac1, Ras-related C3 botulinum toxin substrate 1; MMP2/9/14, matrix metalloproteinase-2, -9, and -14; FoxM1, forkhead box protein M1; PCNA, proliferating cell nuclear antigen; Cyclin, a protein family that controls the progression of a cell through the cell cycle; MKI67, marker of proliferation Ki-67; CDK2, cyclin-dependent kinase-2; CDKN3, cyclin-dependent kinase inhibitor 3; p-Src, phosphorylated proto-oncogene tyrosine-protein kinase; PDGF, platelet-derived growth factor; p-AKT, phosphorylated protein kinase B; p-GSK3β, phosphorylated glycogen synthase kinase 3 beta; p-MDM2, phosphorylated E3 ubiquitin-protein ligase; EGFR, epidermal growth factor receptor; ERα/β, estrogen receptor alpha/beta; FASN, fatty acid synthase; CYP3A4, cytochrome P450 3A4; COX-1/2, cyclooxygenase-1/2; E2F, E2 factor; E2F1, E2F transcription factor 1; KLK3/4, prostate-specific antigen 3/4; Survivin (BIRC5), an inhibitor of apoptosis protein; RBL1, retinoblastoma-like protein 1; VEGF, vascular endothelial growth factor; VEGFR2, vascular endothelial growth factor receptor-2; Cytochrome C, a heme protein localized in the compartment between the inner and outer mitochondrial membranes; PARP, poly (ADP-ribose) polymerase; Caspase-3, an endo-protease which regulates inflammatory and apoptotic signaling networks; p-p53, phosphorylated tumor suppressor p53; Bcl-2, B-cell lymphoma-2 as an apoptotic regulator. Figure adapted from De Silva, S.F. et al. [91].

**Figure 10 ijms-22-04054-f010:**
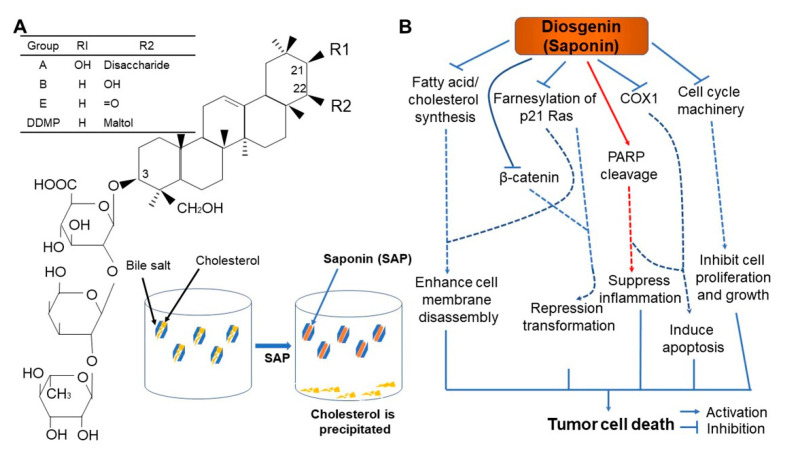
Chemical structure and in vitro cholesterol-lowering mechanism of saponin (**A**), and schematic representation of plausible anticancer mechanism of saponin derivatives at the cellular level (**B**) [95]. p21, a potent cyclin-dependent kinase inhibitor; β-catenin, a core component of the cadherin protein complex; COX, cyclooxygenase; PARP, poly (ADP-ribose) polymerase. →, activation; ⊥, inactivation; ---, indirect activation. Figure adapted from Podolak, I. et al. [95].

**Figure 11 ijms-22-04054-f011:**
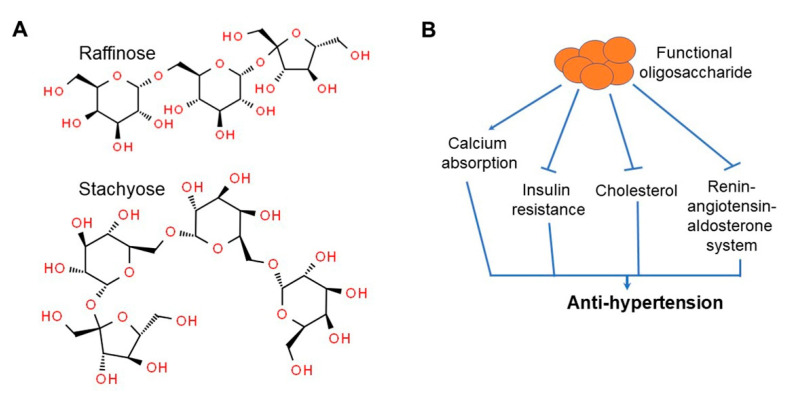
Chemical structures of soybean oligosaccharides comprising stachyose and raffinose (**A**) and effect of functional oligosaccharides on high blood pressure (**B**). Figure adapted from Zhu, D. et al. [132].

**Figure 12 ijms-22-04054-f012:**
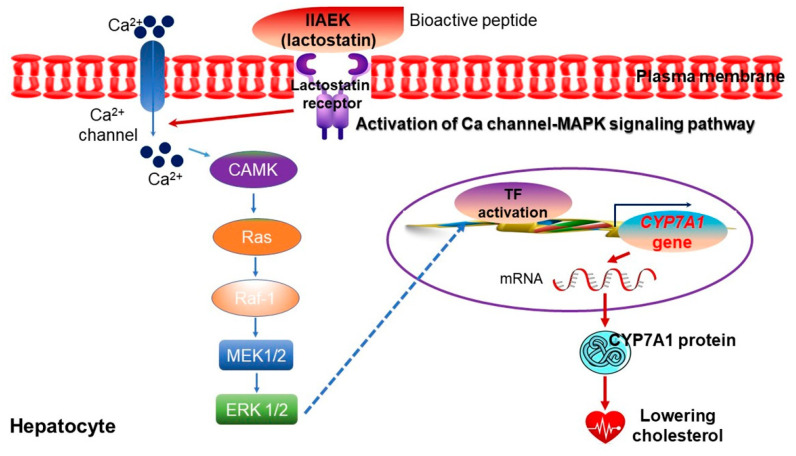
Mechanism of cholesterol degradation through lactostatin (IIAEK)-driven *CYP7A1* expression in hepatic carcinoma HepG2 cell line. CYP7A1, cytochrome P450 monooxygenase; CAMK, calmodulin kinase; ERK, extracellular-signal-regulated kinase; MEK, mitogen-activated protein kinase kinase; Ras, a small GTPase protein; Raf-1, proto-oncogene serine/threonine-protein kinase; TF, transcription factor. Figure adapted from Nagaoka, S. [150].

**Figure 13 ijms-22-04054-f013:**
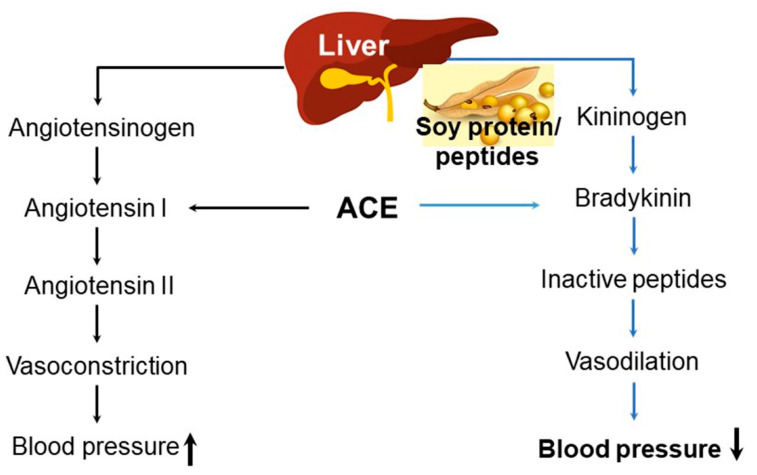
Regulation of angiotensin-converting enzyme (ACE) in blood pressure by soybean protein. ACE catalyzes the degradation of bradykinin, a blood lowering-protein/peptide in the kallikrein–kinin system. Inhibition of ACE is postulated to be an effective medical target in the treatment of hypertension. Hypertension is a key factor associated with several diseases, such as cardiovascular disease and stroke [5]. Figure adapted from Chatterjee, C. et al. [5].

**Figure 14 ijms-22-04054-f014:**
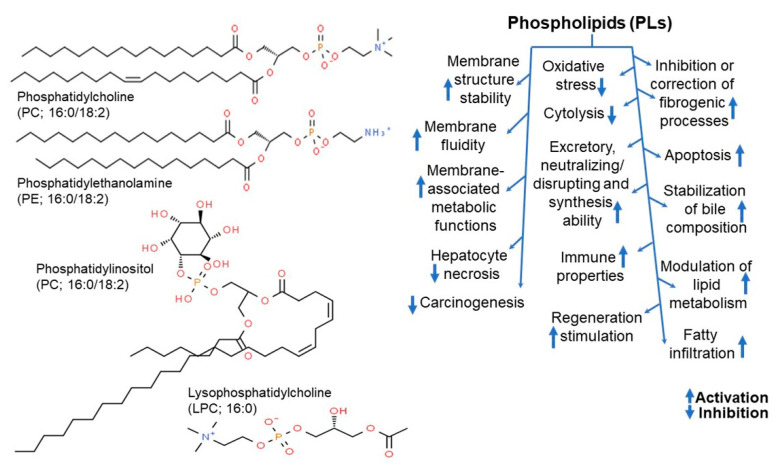
Structure of the major phospholipids (PLs) found in soybean lecithin (**left panel**) and action mode of PLs in liver diseases (**right panel**) [154].

**Figure 15 ijms-22-04054-f015:**
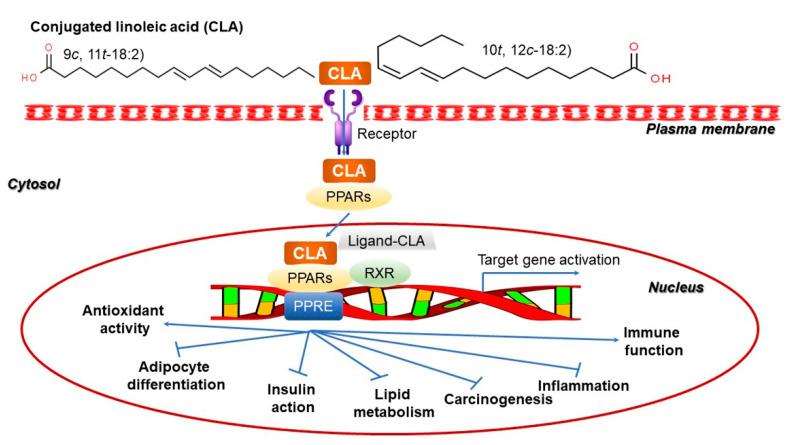
Schematic illustration of the CLA-regulated biological pathway during carcinogenic, adipose, diabetic, and cardiovascular diseases. CLA conjugated to nuclear receptors such as PPARs combines with a counterpart nuclear receptor named RXR to transcriptionally downregulate the target genes related to lipid metabolism including cellular differentiation of adipocytes, cancer cells, inflammatory cells, and pancreatic cells [184]. CLA, conjugated linoleic acid; PPAR, peroxisome proliferator-activated receptor; PPRE, peroxisome proliferator responsive element; RXR, retinoid X receptor. Figure adapted from Yang, B. et al. [184].

**Figure 16 ijms-22-04054-f016:**
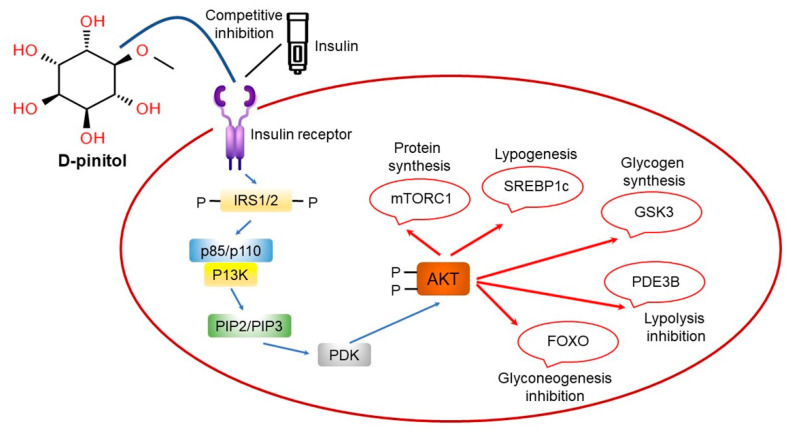
Insulin-resistant effects of pinitol. IRS, insulin receptor substrate; p85/p110, phosphoinositide-3-kinase-alpha subunits; PI3K, phosphoinositide-3-kinase; PIP2/3, probable plasma membrane intrinsic aquaporin protein; PDK, phosphoinositide-dependent protein kinase; AKT, protein kinase B; SREBP1c, sterol regulatory element-binding protein 1; GSK3, glycogen synthase kinase 3; PDE3B, cyclic nucleotide phosphodiesterase 3B; mTORC1, mechanistic target of rapamycin (mTOR) complex 1; FOXO, the O class of the forkhead box class transcription factors. Figure adapted from Antonowski, T. et al. [196].

**Table 1 ijms-22-04054-t001:** Concentration of nutritional components of soybean [1].

Component	Nutritional Value (Per 100 g)	Component	Nutritional Value (Per 100 g)
Carbohydrates	30.20 g	Glycine	1.88 g
Sugars	7.30 g	Proline	2.38 g
Protein	36.49 g	Serine	2.36 g
Tryptophan	0.59 g	Fat	19.94 g
Threonine	1.77 g	Saturated FA	2.89 g
Isoleucine	1.97 g	Monounsaturated FA	4.40 g
Leucine	3.31 g	Polyunsaturated FA	11.26 g
Lysine	2.71 g	Water	8.54 g
Methionine	0.55 g	Vitamin A	0.001 g
Phenylalanine	2.12 g	Vitamin B_6_	0.006 g
Tyrosine	1.54 g	Vitamin C	0.047 g
Valine	2.03 g	Vitamin K	0.277 g
Arginine	3.15 g	Calcium	1.57 g
Histidine	1.10 g	Magnesium	0.28 g
Alanine	1.92 g	Phosphorous	0.704 g
Aspartate	5.12 g	Sodium	1.797 g
Glutamate	7.87 g	Zinc	0.002 g
Total calories		466 kcal	

FA, fatty acid.

**Table 2 ijms-22-04054-t002:** Potential bioactive peptides derived from soybean and its by-products [5,134].

Peptide Sequences	Biological Effects
APP; IPP; AFH; PPYY; PPYY; YVVPK; IPPGVPYWT; LAIPVNKP; LPHF; VLIVP; SPYP; WL; NWGPLV; IVF; LLF; LNF; LSW; IAV; LEF; LEPP; FFYY; FVP; LHPDAQR; VNP; WNPR; WHP; VAHINVGK; YVWK; SY; GY	ACE inhibitor
ADPVLDNEGNPLENGGTYYI	ACE inhibitor and antioxidant
KNPQLR; EITPEKNPQLR; RKQEEDEEQQRE	Fatty acid synthase inhibitor
VRIRLLQRFNKRS	Appetite suppressant
HCQRPR; QRPR	Phagocytosis-stimulating peptide
VK; KA; SY	Lower triglyceride
ILL; LLL; VHVV	Lipolysis
HHL; PGTAVPK; YVVFK; IPPCVPYWT; PNNKPFQ; NWGPLV; TRRVF	Antihypertensive
PGTAVPK; HTSKALDMLKRLGK	Antimicrobial
RQRK; VIK	Anti-inflammatory
IQN	Adipogenesis inhibition
QRPR; HCQRPR	Immunomodulator
LPYP; LPYPR; WGAPSL; VAWWMY; FVVNATSN; IIAEK	Hypocholesterolemic
Vglycin	Antidiabetic
IAVPGEVA; IAVPTGVA;	Hypocholesterolemic and antidiabetic
LLPHH; RPLKPW	Antioxidative and antihypertensive
(X)MLPSWSPW; SLWQHQQDSCRLQLQGVNLFPCEL HIMELIQGRGDDDDDDD	Anticancer
Bowman-Birk inhibitor	Anticancer, protease inhibition, and chemoprevention
SKWQHQQDSCRKQKQGVNLTPCEKHIHEKIQGRGD DDDDDDDD	Antioxidative, anti-inflammatory, anticancer, and hypocholesterolemic
LPYPR; PGP	Antiobesity
YPFVV; YPFVVN; YPFVVNA	Antidiabetic, immunomodulator, lower triglyceride, and suppress feed and intestinal transit

A, Alanine; R, Arginine; N, Asparagine; D, Aspartic acid C, Cysteine; E, Glutamic acid; F, Phenylalanine; K, Lysine; L, Leucine; I, Isoleucine; M, Methionine; Q, Glutamine, G, Glycine, H, Histidine; S, Serine; T, Threonine; W, Tryptophan; Y, Tyrosine; V, Valine.

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
