# Peer review of "Physiologically Active Molecules and Functional Properties of Soybeans in Human Health—A Current Perspective"

_ijms, 2021, doi:10.3390/ijms22084054_

Round 1
Reviewer 1 Report
It is a topic of interest to the scientific community, this is a review article and the lot of numbers of references are included, which is the good side of the presented manuscript. Moreover, the value of the article is increased by the number of figures included in the manuscript which are of great interest.
Author Response
I have attached the two responses.

Reviewer 2 Report
L39: United States
Table 1. I would suggest restructuring Table 1. First, please avoid using red font color as it looks like if it were a correction. Additionally, I would remove A. A.
Figure 1 needs sources. Figure 1 C should be amino acids instead of Protein.
I’m not sure if Figure 2 should be kept as all the information presented on it is written in text. Maybe it could be used as a graphical abstract.
Figure 4. the text in the bottom right corner is hard to read, please improve.
Figure 5. was this figure created by the authors? If not, please cite the source.
The paper is easy to read, however, there are some minor spelling mistakes and sentences that should be improved.
However, I think the authors should write about the fact that soybeans are currently the only GMO beans that are commercially available in the U.S., and there are many countries which do not enable GMO plants. This issue should be at least mentioned and concerns regarding GMO plants should be cited here. Let the reader decide if the risks of being a GMO plant overtake the benefits of soybeans.
Author Response
I have attached the two responses.
